# The Role of the Insular Cortex and Serotonergic System in the Modulation of Long-Lasting Nociception

**DOI:** 10.3390/cells13201718

**Published:** 2024-10-17

**Authors:** Ulises Coffeen, Gerardo B. Ramírez-Rodríguez, Karina Simón-Arceo, Francisco Mercado, Angélica Almanza, Orlando Jaimes, Doris Parra-Vitela, Mareli Vázquez-Barreto, Francisco Pellicer

**Affiliations:** 1Laboratorio de Neurofisiología Integrativa, Instituto Nacional de Psiquiatría Ramón de la Fuente Muñiz, Mexico City 14370, Mexico; coffeen@inprf.gob.mx (U.C.); simonk@inprf.gob.mx (K.S.-A.); orland@inprf.gob.mx (O.J.); 2Laboratorio de Neurogénesis, Instituto Nacional de Psiquiatría Ramón de la Fuente Muñiz, Mexico City 14370, Mexico; gbernabe@inprf.gob.mx; 3Laboratorio de Fisiología Celular, Instituto Nacional de Psiquiatría Ramón de la Fuente Muñiz, Mexico City 14370, Mexico; fmercado@inprf.gob.mx (F.M.); almanza@inprf.gob.mx (A.A.); 4CIANyD Centro Integral Para la Atención de Neuropatía y Dolor, Toluca 50110, Mexico; neuropatiaclinic@gmail.com; 5Servicio de Anestesiología, Hospital General de México, Mexico City 06720, Mexico; mareli_vazquez@hotmail.com

**Keywords:** serotonin, pain, insular cortex, long-lasting nociception

## Abstract

The insular cortex (IC) is a brain region that both receives relevant sensory information and is responsible for emotional and cognitive processes, allowing the perception of sensory information. The IC has connections with multiple sites of the pain matrix, including cortico-cortical interactions with the anterior cingulate cortex (ACC) and top-down connections with sites of descending pain inhibition. We explored the changes in the extracellular release of serotonin (5HT) and its major metabolite, 5-hydroxyindoleacetic acid (5HIAA), after inflammation was induced by carrageenan injection. Additionally, we explored the role of 5HT receptors (the 5HT1A, 5HT2A, and 5HT3 receptors) in the IC after inflammatory insult. The results showed an increase in the extracellular levels of 5HT and 5-HIAA during the inflammatory process compared to physiological levels. Additionally, the 5HT1A receptor was overexpressed. Finally, the 5HT1A, 5HT2A, and 5HT3 receptor blockade in the IC had antinociceptive effects. Our results highlight the role of serotonergic neurotransmission in long-lasting inflammatory nociception within the IC.

## 1. Introduction

The insular cortex (IC) is a brain region that both receives sensory information and is responsible for different emotional and cognitive processes [1,2,3]. It is the locus where sensory information is perceived. The IC is responsible for functions required for basic survival such as pain perception, odor perception, taste recognition and memory, and more complex processes such as emotional awareness and empathy [4,5,6,7,8,9,10]. The neuroanatomic and cytoarchitectonic features of the IC determine its functionality [11,12,13]. Specifically, the IC has connections with multiple sites of the pain matrix, including cortico-cortical interactions with the anterior cingulate cortex (ACC) and top-down connections with sites of descending pain inhibition [14,15,16,17].

Pain processing via the pain matrix involves many neurotransmission systems. For instance, ascending pain transmission via the glutamatergic system contributes to synaptic plasticity in the IC and ACC mediated by NMDA and AMPA receptor activation and, through long-term potentiation, produces long-lasting pain [18,19]. The cholinergic system is implicated in pain memory acquisition, and the inhibitory M2 muscarinic receptor modulates neuropathic pain in the ACC [20,21]. On the other hand, pain relief requires the participation of different inhibitory systems. The activation of the opioid, GABA, glycine, and cannabinoid systems in the ACC and IC modulates the response to acute pain and alleviates neuropathic pain [22,23,24,25,26,27]. Additionally, catecholaminergic transmission plays an important role in the modulation of long-term nociceptive hypersensibility in the IC after inflammatory or neuropathic injury.

The role of monoamines in inflammatory and neuropathic pain has been described extensively [28,29,30]. One of the more intricate monoamines derived from tryptophan is the neurotransmitter serotonin (5-hydroxytryptamine [5HT]), which is related not only to nociception but also to many other functions, including intestinal movement, vasoconstriction, mood, appetite, sleep, anxiety, and depression. The complexity of 5HT function is due to the large variety of 5HT receptor subtypes (the 5HT1, 5HT2, 5HT3, 5HT4, 5HT5, 5HT6, and 5HT7 receptors) and on the differences in localization of these receptors.

In the periphery, 5HT is part of the so-called “inflammatory soup”, which also contains other mediators such as histamine, prostaglandins, and bradykinin, and contributes to the induction and maintenance of nociceptive pathway hypersensitivity [29,31].

In the spinal cord (SC), low-frequency (4 Hz) but not high-frequency transcutaneous electric nerve stimulation in a rat joint inflammation model is capable of increasing the release of 5HT, resulting in an antinociceptive effect [32]. Additionally, serotonin has been reported to have dual effects on formalin-induced pain in the rat SC; the intrathecal administration of a low dose of 5HT exerts antinociceptive effects, while a high dose of 5HT produces pronociceptive effects, possibly through 5HT1A receptor activation [33].

At the supraspinal level, there is less information about the pathophysiological relevance of the monoaminergic system to pain processing. Exposure to a novel environment produces antinociceptive effects in rats injected with formalin, and this antinociceptive effect is correlated with lower extracellular levels of 5-hydroxyindoleacetic acid (5HIAA, the main metabolite of 5HT) and DOPAC (a metabolite of dopamine) in the medial prefrontal cortex [34]. Additionally, the basal levels of 5HT decrease in the ventromedial thalamus in a neuropathic pain model induced by the ligature of the L5 and L6 spinal roots [35]. In contrast, 5-HIAA levels are increased in both the periaqueductal grey matter and in the trigeminal nucleus after peripheral nerve axotomy [36].

As mentioned above, the IC is related to the development and perception of inflammatory and neuropathic pain [1,37,38,39,40]. In rodents, 5HT fibers from the caudal and rostral dorsal raphe spread extensively throughout the IC and are distributed in all layers of this cortical region [41]. The 5HT1A and 5HT2A receptors and, to some extent, the 5HT3 receptor, are highly expressed in the PFC and are localized to both pyramidal cells (PCs) and interneurons (Ins) [42,43]. The activity of the 5HT1A and 5HT2A receptors in the anterior IC have been documented in different experimental models in humans and nonhuman primates [44,45,46].

However, the pain-related role of the serotonergic system in the IC has not been fully studied. Thus, we explored the changes in the extracellular release of 5HT and 5HIAA (the main metabolite of 5HT), as well as the protein levels of the 5HT1A, 5HT2A, and 5HT3 receptors in the IC, in an inflammatory pain model. Furthermore, we explored the effects of pharmacological antagonism of 5HT receptor on the response to painful stimuli in the context of inflammatory pain.

## 2. Materials and Methods

### 2.1. Animals

The experiments were conducted in accordance with the ethical regulations of the International Association for the Study of Pain [47] and in compliance with Mexican Official Norm NOM-062-ZOO-1999 on laboratory animal research, and under the approval of both the research ethics committee (CEI/C/016/2022) and the internal committee for the care and use of laboratory animals (CICUAL/01/2022) of the Ramón de la Fuente Muñiz National Institute of Psychiatry. The number of animals used in this study was the minimum necessary to achieve statistical significance.

Male Wistar rats (250–300 g) were maintained in our institution’s animal facility. The animals were kept in individual transparent acrylic cages on a 12:12 h light–dark cycle at 23 °C and 52% humidity and provided ad libitum access to food and water. To reduce stress, the rats underwent 20 min habituation sessions in experimental acrylic cages for five consecutive days. For all surgeries, the rats were anaesthetized with 2% isoflurane.

#### Experimental Design

The main goal of the study was to ascertain the role of the serotonergic system within the IC in the context of pain caused by an inflammatory process. The inflammatory process was induced using the intraplantar carrageenan injection model. The aforementioned method was applied to all experimental series. Furthermore, two nociceptive tests were employed to ascertain the behavioral response to disparate types of stimulation.

The experimental design comprised two principal sections. The initial phase of the study entailed the measurement of the physiological or pathophysiological response elicited by CAR injection. This section was divided into two experimental series. In the initial series of experiments, the extracellular release of 5HT and its metabolite 5HIAA was measured in the IC under a range of experimental conditions. In the second series of experiments, the expression of 5HT1A, 5HT2A, and 5HT3 receptors in the IC was measured. The second part of the study aimed to ascertain whether pharmacological manipulation could induce a change in the behavioral response to nociceptive tests. To this end, specific antagonists for each receptor type were administered directly into the IC. The following sections provides a detailed account of the aforementioned procedures.

### 2.2. Inflammation Induction

Inflammation was induced by the injection of lambda carrageenan (Sigma Chemical Co., St. Louis, MO, USA, 1% in saline solution, 100 μL) into the right hind paw.

### 2.3. Physiological Experiments

#### In Vivo Microdialysis

The rats were anaesthetized with 2% isoflurane mixed with 98% O_2_ and mounted in a stereotaxic frame. A guide cannula (CMA-11-Microdialysis, Acton, MA, USA) was stereotaxically implanted into the IC. Forty-eight hours after cannulation, a microdialysis probe (SciPro Inc. (Austin, TX, USA) 12, 2 mm tip length) was inserted into the guide cannula so that its tip was in the rostral agranular insular cortex (RAIC) (AP = 1 mm from bregma, ML = 4.8 mm, DV = −5.8 mm from the meninges) [48].

Sterile artificial cerebrospinal fluid (aCSF) (145 mM NaCl, 2.8 mM KCl, 3.0 mM CaCl_2_, and 5.4 mM D-glucose (pH 7.2)) was continuously perfused through the cannulas at a rate of 2 μL/min using a micro-infusion pump (KD Scientific, Holliston, MA, USA). The animals were individually housed for the duration of the experiment, and microdialysate samples were collected from the freely moving animals at 20 min intervals for 180 min in microvials containing 4 μL of 0.08% L-glutathione (Sigma-Aldrich, Merck KGaA, Darmstadt, Germany) to prevent the oxidation of monoamines.

### 2.4. Biochemical Analysis

Isocratic high-performance liquid chromatography with electrochemical detection (HPLC Waters 2695 and ECD Waters 2465) was used to quantify 5HT and 5HIAA levels. A mobile phase consisting of 95% 12.5 mM citric acid, 0.07 mM 1-octanesulfonic acid sodium salt, 0.05 mM EDTA, 25 nM ortho phosphoric acid, and 5% methanol (adjusted to pH 3.2 with 10 M KOH) was injected at a rate of 0.1 mL/min through an X Terra C 18 (2.1 Å ~50 mm, 3.5 μM ODS) column. Online data capture was performed using Waters Empower software version 1154, for HPLC.

The animals were divided into the following groups: the control group (saline solution, n = 8), in which the basal extracellular concentrations of 5HT and 5HIAA were measured for three consecutive hours; the mechanonociceptive group (n = 8), in which the extracellular concentrations of 5HT and 5HTIAA were measured over a period of 3 h during the delivery of mechanonociceptive stimuli (basal, 1 h and 3 h; mechanonociceptive test, vide infra); and the inflammation group (n = 8), in which the extracellular concentrations of 5-HT and 5-HIAA were measured prior to the induction of inflammation and over a period of three hours after inflammation induction during the delivery of mechanonociceptive stimuli (basal, 1 h and 3 h).

### 2.5. Western Blotting

Brain tissues (control and CAR groups, n = 11 per group) were dissected and lysed with RIPA buffer (1 × PBS, 0.1% SDS, 1% NP40, 0.5% sodium deoxycholate, 0.24 mg/mL AEBSF, 8 μg/mL aprotinin, 10 μg/mL leupeptin, 4 μg/mL pepstatin, 5 mM benzamidine, 20 mM β-glycerophosphate, 10 mM NaF, 1 mM Na_3_VO_4_, 1 mM EDTA, and 1 mM EGTA). The brain tissues were homogenized with an ultrasonic homogenizer. The total protein content was quantified using a Bradford assay kit (Bio-Rad, Hercules, CA, USA). The proteins were separated and transferred to a nitrocellulose membrane. The membrane was probed with a rat anti-5HTR1A antibody, a rat anti-5HTR2A antibody, and a rat anti-5HT3a receptor antibody (1:1000; Boster Biological Technology, Pleasanton, CA, USA). GAPDH was used as a loading control (1:1000; mouse anti-GAPDH antibody, Abcam; Cambridge, MA, USA). The proteins were visualized with a Millipore-enhanced chemiluminescence detection system (Naucalpan, Estado de México, México) and a ChemiDoc Touch System from Bio-Rad. After identifying every protein, membranes were stripped and exposed to the enhanced chemiluminescence detection system to ensure that the previous reaction was eliminated. Densitometric analysis was performed with the Image Lab software version 6.1 (Bio-Rad).

### 2.6. Pharmacological Experiments

#### Cannula Implantation and Microinjection

Two days before the experiment, a 17 mm long guide cannula (inner diameter: 21 G) was stereotaxically inserted into the left RAIC. The guide cannula tip was placed over the meninges (AP: 1.0 from bregma, ML: 4.8, DV: −5.2 from the meninges). All animals were allowed to recover for 48 h [48].

On the day of the experiment, a cannula (28G, Small Parts Inc., Logansport, IN, USA) that was 0.2 mm longer than the guide cannula was inserted into the guide cannula. The cannula was connected by polyethylene tubing (PE-10, 20 cm long) to a 25 μL syringe (Hamilton Co., Reno, NV, USA) filled with different drugs (vide infra) or vehicle (control group, n = 10). For all animals, 2 μL of solution was injected under the control of a syringe pump at a rate of 0.5 μL/min (Harvard Apparatus, MA, USA). The cannula was left in place for an additional 60 s to reduce the chance of reflux. These microinjection parameters have been reported not to induce tissue damage [49]. The drug dosages used in this work were chosen based on previous reports demonstrating their pharmacological efficacy [50,51].

Immediately after IC microinjection, the rats were placed in an aesthesiometer and plantar test devices (to measure mechanical and thermal nociception, *vide infra*). Different groups of animals were administered different drugs (5HT, 5HIAA, or 5HT1A, 5HT2A and 5HT3 receptor antagonists (n = 10, 2 μL each)).

### 2.7. Drugs

The drugs used in the different experiments are summarized in Table 1.

#### Nociceptive Tests

Mechanical nociception was measured using an aesthesiometer (Von Frey mechanonociception; Ugo Basile Gemonio (VA), Italy, mod 37400-001). In this test, the force threshold of metallic filaments required to produce a reflex upon application to the hind paw was determined (in grams [g]; 10 g intervals from 0 to 50 g). Four measurements for the contralateral hind paw were averaged to obtain the values for each time point (1 h and 3 h after intraplantar CAR injection).

Thermal nociception was measured using radiant heat in the plantar test (Ugo Basile, model 7370) according to the Hargreaves method. The right paw withdrawal latency (PWL) was determined to the nearest 0.1 s using the electronic timer in the plantar test apparatus. The cut-off time was 20 s to avoid tissue damage. The average of three trials per hind paw was calculated for each time point.

### 2.8. Histological Verification

At the end of the experiment, the placement of the microdialysis probe was verified. Briefly, the animals were intracardially perfused with physiological saline solution, followed by 10% formaldehyde. The brains were postfixed for 2 days and cut into 40 μm coronal slices, which were immediately placed on glass slides and imaged with a scanner (HP Scanjet 5550C). The placement of the probe was analyzed by comparison to an anatomical atlas [48]. Animals in which the injection site was outside the IC were excluded (Appendix A).

### 2.9. Oedema Size Measurement

To verify whether carrageenan injection produced plantar inflammation, the oedema size was measured using a digital plethysmometer (Ugo Basile, 37140). The paws of the animals were submerged below the calcaneal-tibial joint, and volume displacement was recorded 3 times for each rat. Immersion of the paw in water resulted in water displacement, and a signal was sent to the digital display indicating the volume displacement (0.01 mL resolution). The mean value for each rat was obtained (Appendix A).

### 2.10. Statistical Analysis

For microdialysis, the results were analyzed using a repeated measures ANOVA, followed by Tukey’s post hoc test. Pearson’s correlation analysis was performed to test whether there was a relationship between 5HT and 5HIAA release and nociceptive stimulation. For pharmacological experiments, the differences between groups were analyzed with two-way ANOVA followed by Tukey’s multiple comparisons post hoc test using GraphPad Prism version 9.0.1 for macOS (GraphPad Software, San Diego, CA, USA; www.graphpad.com). For all the statistical analyses, *p* ≤ 0.05 indicated statistical significance.

For Western blot analysis, the results were analyzed using SigmaStat 3.1 software (Systat Software Inc., San Jose, CA, USA). The data are presented as the means ± SEMs. The results were analyzed using a one-way ANOVA, but when the normality test failed, they were analyzed with a Kruskal–Wallis one-way ANOVA on ranks, both followed by Tukey’s post hoc test. In the latter, we report the q value, which is used to evaluate the difference between group means. Additionally, we indicate the H value, which is the test statistic for the Kruskal–Wallis test and the degrees of freedom (d.f.).

## 3. Results

### 3.1. Physiological Experiments

#### 3.1.1. Inflammation Increases Concentrations of 5HT and 5HTIAA

The concentrations of 5HT and 5HTIAA during the inflammatory process induced by intraplantar injection of carrageenan (inflammation group, n = 8) were greater than the basal concentrations and those in the control (vehicle, n = 8) and mechanical nociception groups (n = 8). The extracellular concentration of 5HT began to increase approximately 20 min after CAR injection and remained high until reaching a difference of 52% at 180 min with respect to the control group and the mechanonociception group (ANOVA, F = 40.39, *p* = 0.0001) (Figure 1A), while the extracellular levels of 5HIAA began to increase from 80 min after CAR injection, reaching a difference of 70% at 180 min with respect to the other groups (ANOVA, F = 17.89, *p* = 0.0008) (Figure 1B). Moreover, there was a negative correlation between the increase in 5-HIAA release in the IC and the decrease in the paw withdrawal threshold (PWT) in the inflammation group (Pearson’s correlation, r = −0.692, *p* < 0.005) (Figure 1C). The mechanonociception group, which only received acute mechanical stimulation at different times, showed no difference in both extracellular 5HT and 5HIAA levels compared to the control group.

#### 3.1.2. Protein Levels of the 5HT1A Receptor Shows a Sustained Increase after Inflammation

The protein levels of the 5HT1A receptor in the IC significantly increased after 3 h post-carrageenan administration compared to those in control rats (q = 3.64, *p* < 0.05). Interestingly, the protein levels of the 5HT1A receptor remained higher 24 h after carrageenan treatment than in control rats (q = 3.32, *p* < 0.05) (Figure 2, H = 8.15, d.f. = 2, *p* = 0.017). However, although both show an upward trend, there were no statistical differences in the levels of the 5HT2A (Figure 2, H = 5.79, d.f. = 2, *p* = 0.055) and 5HT3A (F2,32 = 0.59, *p* = 0.55) receptors at 3 or 24 h after carrageenan administration (Figure 2).

### 3.2. Pharmacological Experiments

#### 3.2.1. Blocking 5HT3R Decreases Mechanical Nociception

The PWT to mechanical stimulation was diminished 1 h after carrageenan injection in all groups (n = 10 per group) compared to the basal values (ANOVA, *p* = 0.0001). Microinjection of ondansetron (a 5HT3 receptor antagonist) had a significant antinociceptive effect 1 h after carrageenan injection compared to the control, 5HT, and 5HIAA groups (*p* = 0.0492; *p* = 0.0274; and *p* = 0.0317, respectively). Additionally, the nociceptive effect was diminished in the 5HT3 group compared to the 5HT group 3 h after CAR injection (*p* = 0.0038) (Figure 3).

#### 3.2.2. Blocking 5HT2AR and 5HT3R Decreases Thermal Nociception

In the thermal nociceptive test, inflammation induced a decrease in the PWL in all groups (n = 10 per group) 1 h after carrageenan injection (ANOVA, *p* = 0.0001). Microinjection of 5HT1A, 5HT2A, and 5HT3 receptor antagonists partially increased the PWL. There was a significant difference in the PWL in the 5HT2A group with respect to control (*p* = 0.0034) and 5HT (*p* = 0.0019) groups. Additionally, at 3 h, the 5HT2A group showed a statistically antinociceptive effect when compared to the 5HT group (*p* = 0.040). The 5HT3 group showed a significant difference with respect to the control (*p* = 0.0148) and 5HT (*p* = 0.0046) groups one hour post-CAR injection. Moreover, 3 h after injection, the 5HT3 group showed a significantly antinociceptive effect when compared to the control (*p* = 0.0312) and 5HT groups (*p* = 0.0004). (Figure 4).

## 4. Discussion

The present work highlights the role of the serotoninergic system in the IC in long-lasting inflammatory pain. Our results revealed an increase in the extracellular level of 5HT and its major metabolite 5-HIAA during inflammation compared to physiological levels and in the absence of nociceptive stimulation. Additionally, inflammation increased 5-HT1A receptor expression in the IC. Moreover, blockade of 5HT1A, 5HT2A, and 5HT3 receptors in the IC exerted an antinociceptive effect.

### 4.1. Physiological Experiments

Our results are consistent with those of Zhang et al.’s report. The authors reported that the extracellular concentrations of 5HT and 5-HIAA in the SC dorsal horn and periaqueductal grey (PAG) significantly increase following carrageenan-induced inflammation [52]. Moreover, as mentioned earlier, 5HT and 5HIAA levels within the PAG and the trigeminal nucleus increase after peripheral nerve axotomy [36]. However, 5HT and 5HIAA levels decrease in the ventrobasal thalamus and medial prefrontal cortex, respectively, in neuropathic pain and formalin-induced pain models [34,35].

The differences in these results may be due to the brain region studied, the pain model used, or even the technique employed to measure the release of the neurotransmitter. The measurement technique used in the present work (microdialysis in freely moving rats) allowed us to quickly measure the release of neurotransmitters while the animals moved freely. Moreover, we were able to record nociceptive responses during the measurement process; this contrasts with previous reports in which brain homogenates were used for measurement, allowing only indirect measurement of brain activity postmortem.

An interesting result of the present study was the increase of up to 70% in the level of the 5HT metabolite 5HIAA during the inflammatory process. This increase correlated with the increase in pain-like behavior in response to mechanical stimulation. A previous study by Chen et al. [53], who used an inflammatory pain model (induced by intraplantar injection of complete Freund’s adjuvant, CFA), showed that exogenous administration of 5HIAA increases thermal hyperalgesia in 5HT transporter KO mice (absence of thermal hyperalgesia) and in wild-type mice pretreated with p-CPA (a competitive 5HT biosynthesis inhibitor). These findings, together with our results, support the assumption that 5HIAA is involved in long-lasting nociception at different levels of the CNS.

In this regard, the increase in the extracellular concentration of 5HT starts to be significant almost from the beginning of the inflammatory process, whereas the increase in 5HIAA is not significant until after one hour. This discrepancy may be attributed to the fact that the inflammatory process initially results in the release of serotonin into the extracellular space, which is subsequently metabolized by MAO to form 5HIAA. The process depends on different factors and does not necessarily follow a specific catabolism time. For instance, Bulat and Supé showed that in entire brain tissue homogenates, 5HT can be transformed into 5HIAA in less than ten minutes [54]. In contrast, Zhang et al. found that following CAR injection, there is a significant increase in 5HT release in the spinal cord after 60 min, while the significant increase in 5HIAA release occurs after 120 min [52].

The effect of 5HT also depends on the physiological/pathophysiological status of the animal. For example, in healthy animals, spinal 5HT mainly exerts an inhibitory influence on pain signaling mechanisms, whereas in animals with lesions in the periphery and/or in the CNS, the bulbospinal 5HT system may exacerbate pain [55]. Our findings suggest that the extracellular levels of 5HT and 5HIAA in the IC remain relatively stable during the acute nociceptive process, even in the presence of repeated mechanical stimulation. However, there seems to be an increase during the inflammatory process. These findings are analogous to those of a previous study examining the dopaminergic system in the IC, which also demonstrated that acute stimulation (thermociception) does not result in any change in extracellular dopamine release [56]. This may be attributed to the nature of the response to acute stimulation, which occurs at the spinal cord level and not in the higher nuclei.

Furthermore, the complex mechanism of 5HT could be related to the expression of multiple 5HT receptors both in the periphery and in the CNS. The IC receives serotonergic inputs [41], and the major serotonin receptor subtypes expressed in this region are the metabotropic 5HT1A and 5HT2A receptors [42] and the ion channel 5HT3 [43]. In this work, we hypothesized that the nociceptive process could alter the protein expression of these receptors and that pharmacological manipulation may exert differential effects on the modulation of nociception.

We observed the overexpression of the 5HT1A receptor 3 and 24 h after inflammation. It was previously shown that the induction of inflammation activates neurotransmission within the IC. Earlier, we showed changes in the release of dopamine in the IC, as well as the upregulation of dopamine D2 receptor mRNA and downregulation of dopamine D1 receptor mRNA, after nociception [56]. This implies an increase in inhibitory receptor expression and a decrease in excitatory receptor expression.

There is evidence of the upregulation of the presynaptic 5-HT1A receptor in the SC in a murine model of diabetic neuropathic pain [57]. Furthermore, Zhang et al. showed that following carrageenan-induced inflammation, the mRNA expression of the 5HT2A receptor in the ipsilateral dorsal horn, bilateral raphe magnus nucleus, ventrolateral PAG, and dorsal raphe nucleus significantly increases, peaking at 3 h [58,59]. Additionally, the exogenous administration of 5HT increases the expression of the 5HT2A receptor after L5 nerve root damage in rats in a dose-dependent manner [60].

We found an increase in 5HT levels after nociceptive stimulation (up to 52% at 3 h after carrageenan injection), which was associated with an increase in extracellular 5HT levels and the overexpression of the 5HT1A receptor (also observed at 3 h after carrageenan injection).

There is also an increase in the expression of the 5HT3 receptor in the SC in myofascial pain syndrome; this increase can be inhibited by thermal therapy [61]. Moreover, Cristidis et al. [62] showed that the 5HT3A receptor is highly expressed in human masseter and tibialis muscles in patients with chronic myofascial temporomandibular disorders compared to pain-free controls. We found a similar expression in 5HT3A receptor expression in rats treated with carrageenan.

### 4.2. Pharmacological Experiments

The three 5HT receptor subtypes showed increased expression, although they responded differently to pharmacological treatment. We found that compared to both the control group and the group that received exogenous 5HT, the group in which the 5HT2A and 5HT3 receptors were inhibited showed significantly reduced nociceptive behavior in the thermal nociception test. However, blocking the 5HT1A receptor did not clearly produce this effect. This makes sense given the variations in the localization of the various receptors and their modes of activation.

The 5HT1A receptor is a G protein-coupled receptor. Its activation reduces the intracellular concentrations of cAMP, opens K^+^ ion channels, and closes Ca^2+^ channels, inhibiting neuronal firing [63]. Furthermore, this receptor is present at presynaptic and postsynaptic sites. The activation of autoreceptors (presynaptic) decreases the release of 5HT in limbic regions. In the hippocampus and cortex, 5HT1A activation increases the GIRK current, leading to hyperpolarization. Additionally, intracerebral administration of 5HT preferentially activates autoreceptors to produce nociceptive effects (for a review, see [64]).

In our study, the exogenous administration of 5HT (acting as an endogenous agonist), as well as the increase in 5HT release caused by the inflammatory process, did not seem to activate 5HT1A autoreceptors. Although selective blockade of 5HT1A with NAN-190 apparently has an antinociceptive effect against thermal stimulation, the 5HT1A receptor does not seem to have a clear role in the induction or maintenance of painful conditions. One possibility is that this antagonist could exert its effect both pre- and postsynaptically, leading to a difference in the desired effect [65].

On the other hand, we found that blocking the 5HT2A receptor produced an antinociceptive effect in response to thermal stimulation. The 5HT2A receptor activates phospholipase C (PLC) through Gq, leads to the accumulation of IP3 and diacylglycerol (DAG) and the activation of protein kinase C (PKC), and causes the release of calcium from intracellular endoplasmic reticulum stores. This receptor is widely distributed throughout the CNS [66]. In accordance with our results, selective blockade of this receptor through the administration of an antagonist via different routes (local, systemic, or intrathecaladministration) can exert antinociceptive effects in different animal models of long-term pain, as well as analgesic effects, as observed in clinical practice [67].

Furthermore, the 5HT2A receptor is expressed mainly in glutamatergic projection neurons (approximately 70%) within the IC and is present in the majority of insular–amygdalar and insular–lateral hippocampal projection neurons, both of which are involved in pain modulation (73–82%) [68]. Moreover, there is evidence that glutamatergic synapses from the IC to the basolateral amygdala encode observational pain in mice [69].

We found that in freely moving rats, microinjection of ondansetron (a 5HT3 antagonist) into the IC reduced not only thermal nociception but also nociception induced by a mechanical stimulus. The 5HT3 receptor is structurally and functionally distinct from the other six 5HT receptor subtypes, whose actions are mediated via G proteins. The 5HT3 receptor is a cation-selective ion channel capable of mediating fast excitatory neurotransmission in the central and peripheral nervous systems by rapidly activating and desensitizing inwards currents, which are carried primarily by Na^+^ and K^+^ ions, and is permeable to Ca^2+^ and other small organic cations [70,71].

The pronociceptive role of the 5HT3 receptor in different areas of the nervous system is well documented. In primary afferent neurons, the 5HT3 receptor mediates sustained and transient effects related to itch and pain behavior in rats [72]. In the rostral ventromedial medulla, the blockade of the 5-HT3 receptors with ondansetron strongly inhibits the mechanical and, to a lesser degree, thermal stimulation-evoked responses of wide-dynamic-range neurons in laminae (V–VI) of the SC in neuropathic model rats [73]. The intrathecal administration of ondansetron reverses mechanical and cold hypersensitivity in a model of chemotherapy-induced neuropathy [74]. Furthermore, in an FMRI study in patients with fibromyalgia, local injection of another 5HT3 receptor antagonist, tropisetron, was found to significantly reduce activation of the contralateral primary somatosensory cortex, contralateral posterior insula, and ACC [75]. Using a single microinjection of ondansetron directly into the IC, we were able to produce an antinociceptive response following inflammation induction, which further supports the pronociceptive function of the 5HT3 receptor.

## 5. Conclusions

Serotonergic neurotransmission in the IC increases during the inflammatory process, and this has a generally pronociceptive effect. The increases in the extracellular concentrations of 5HT and 5HIAA are accompanied by the upregulation of the 5HT1A receptor. Additionally, the blockade of 5HT2A and 5HT3 receptors has an antinociceptive effect. The IC seems to play a pronociceptive role during an inflammatory insult. These phenomena highlight the role of serotonergic neurotransmission within the IC in the development of long-lasting painful conditions.

## Figures and Tables

**Figure 1 cells-13-01718-f001:**
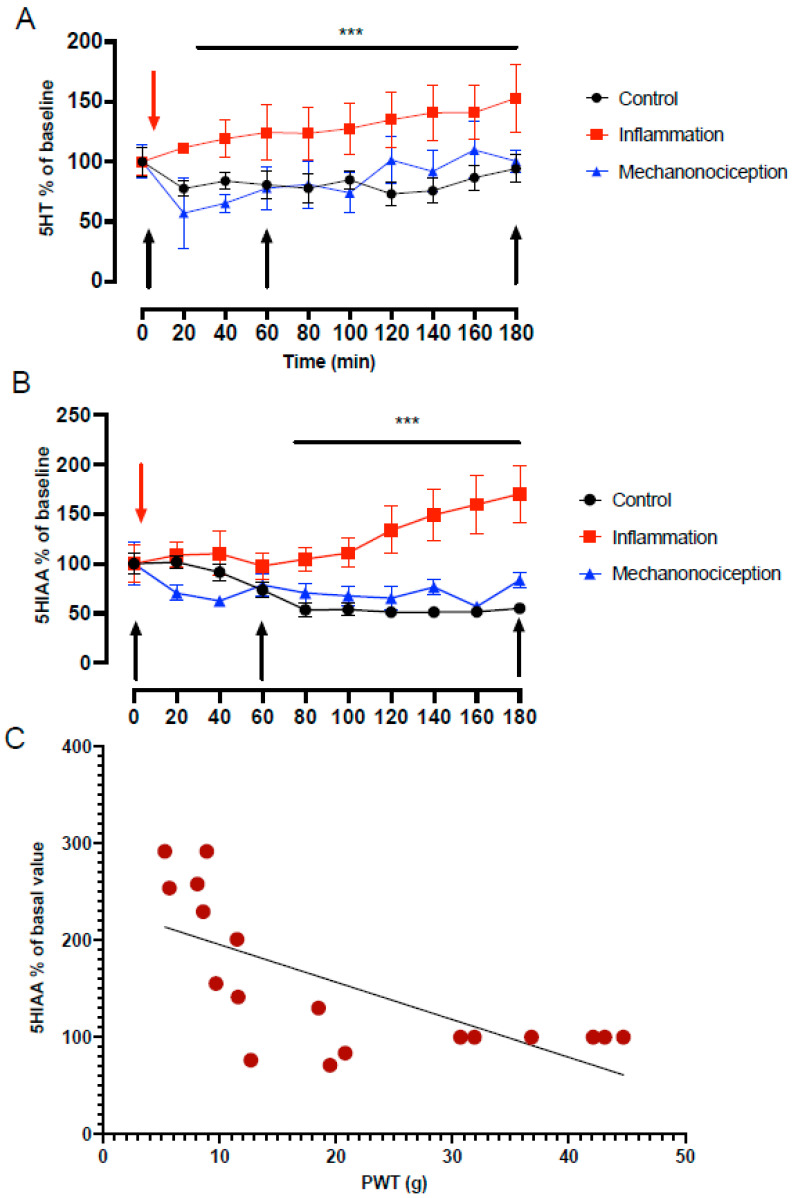
Time course of extracellular levels of 5HT (**A**) and 5HIAA (**B**), expressed as the percentage change, in the insular cortex (IC) of the rat. The red arrow indicates the intraplantar injection of carrageenan (CAR) in the inflammation group. The black arrows indicate the time points of mechanonociceptive stimulation. Values are the mean ± SEM of 8 rats per group. (**A**) Note the progressive and significant increase in the 5HT concentration after CAR injection in the inflammation group compared with control and mechanonociception groups (*** ANOVA, F = 40.39, *p* = 0.0001). (**B**) 5HIAA levels increased 70% at 3 h post CAR injection (*** ANOVA, F = 17.89, *p* = 0.0008). (**C**) There was a negative correlation between the paw withdrawal threshold (PWT) and the percentage change in extracellular 5HIAA levels in the IC (Pearson’s correlation, r = 0.692, *p* < 0.005).

**Figure 2 cells-13-01718-f002:**
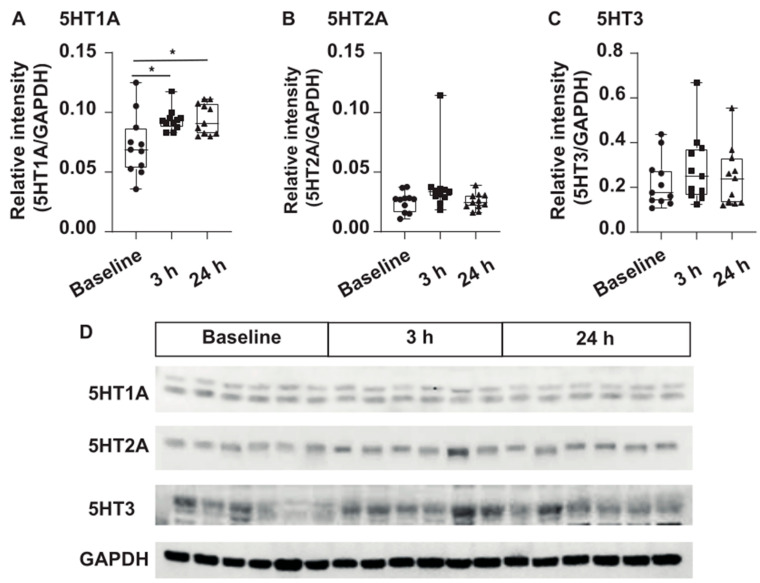
Protein levels of serotonin receptors quantified in the insula cortex (IC) after 3 and 24 h post-induction of inflammation. (**A**) Protein levels of 5HT1A showed an increase after 3 h post-induction of inflammation (*p* < 0.001) that was sustained after 24 (*p* < 0.001) hours post-carrageenan (CAR) administration (F2,15 = 22.93, *p* < 0.001). However, the protein levels of 5HT2A (**B**; F2,16 = 0.32, *p* = 0.72) and 5HT3A (**C**; F2,17 = 0.95, *p* = 0.4) receptors did not show statistically significant differences. Results represent the mean ± the standard error of the mean. Data were analyzed using a one-way ANOVA or Kruskal–Wallis test, both, followed by Tukey’s post hoc test. Differences were considered statistically significant at *p* ≤ 0.05 (*, asterisks). N = 33, n = 11 per group. (**D**) Immunoblots of the 5HT1A, 5HT2A, and 5HT3 receptors are shown. GAPDH was used as a loading control. Blots include proteins of the control group, 3 and 24 h post-induction of inflammation, respectively. Complete immunoblots are displayed in the Appendix A.

**Figure 3 cells-13-01718-f003:**
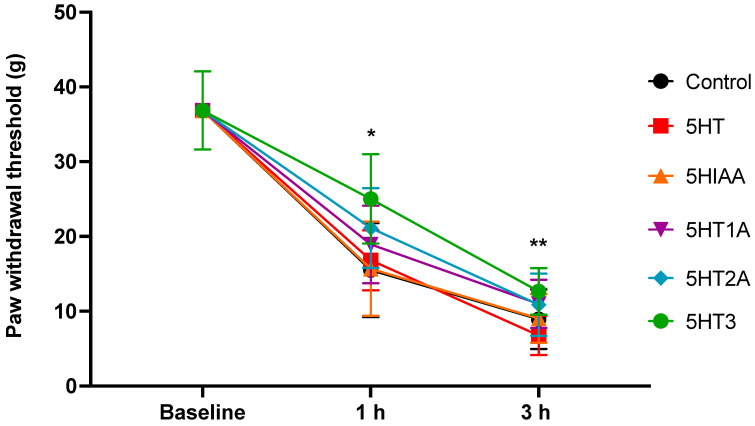
Paw withdrawal threshold (PWT) before and after plantar injection of CAR expressed in grams. The different groups were microinjected into the insular cortex with vehicle (control) or different serotonin-related drugs (5HT, 5HIAA, or antagonists of the 5HT1A, 5HT2a and 5HT3 receptors). Values are the mean ± SEM of 10 rats per group. The microinjection of antagonist of 5HT3 receptors (ondansetron) induced a significant increase in the PWT 1 h post-CAR injection when compared to control, 5HT, and 5HIAA groups (* *p* = 0.0492; *p* = 0.0274; and * *p* = 0.0317, respectively). Additionally, the nociceptive effect was diminished in the 5HT3 group compared to the 5HT group 3 h after CAR injection (** *p* = 0.0038).

**Figure 4 cells-13-01718-f004:**
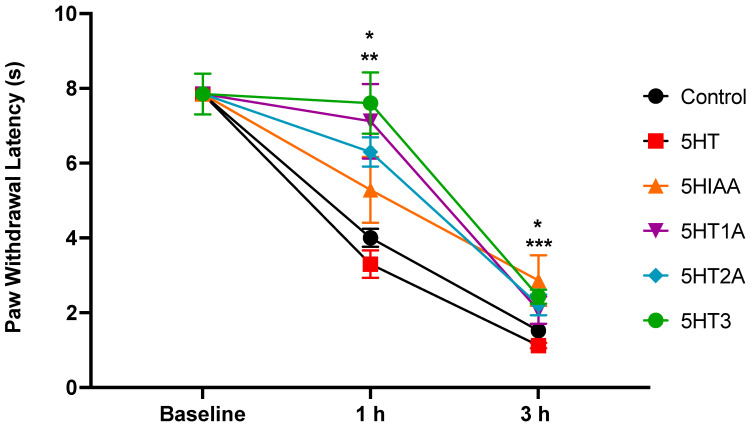
The figure shows the paw withdrawal latency (PWL) before and after plantar injection of CAR expressed in seconds. The different groups were microinjected into the IC with vehicle (control) or different serotonin-related drugs (5HT, 5HIAA, or antagonists of the 5HT1A, 5HT2A, and 5HT3 receptors). Note the significant increase in the PWL in 5HT2A (blue), at 1 h after CAR when compared to control (** *p* = 0.0034) and 5HT (** *p* = 0.0019) groups, and at 3 h, when compared to the 5HT group (* *p* = 0.040). The 5HT3 group (green) showed a significant difference with respect to the control (* *p* = 0.0148) and 5HT (** *p* = 0.0046) groups at 1 h after CAR. Moreover, 3 h after CAR, the 5HT3 group showed a significantly antinociceptive effect when compared to the control (* *p* = 0.0312) and 5HT groups (*** *p* = 0.0004).

**Table 1 cells-13-01718-t001:** The drugs used in the study, their common name, and their function or pharmacological activity.

Function or Pharmacological Activity	Company	Common Name	Chemical Name
Neurotransmitter	Sigma-Aldrich, Merck KGaA, Darmstadt, Germany	Serotonin	3-(2-Aminoethyl)-5-hydroxyindole hydrochloride
Major serotonin metabolite	Sigma-Aldrich, Merck KGaA, Darmstadt, Germany	5-HIAA	5-Hydroxyindole-3-acetic acid
5-HT1A receptor antagonist	Tocris Bioscience, San Jose, CA, USA	NAN-190	1-(2-Methoxyphenyl)-4-(4-phthalimidobutyl)piperazine hydrobromide
Selective 5HΤ3 receptor antagonist	Sigma-Aldrich, Merck KGaA, Darmstadt, Germany	Ondansetron	1,2,3,9-Tetrahydro-9-methyl-3-[(2-methyl-1H-imidazol-1-yl)methyl]-4H-carbazol-4-one hydrochloride
Potent and selective 5-HT_2A_ antagonist; increases the release of dopamine by the medial prefrontal cortex	Sigma-Aldrich, Merck KGaA, Darmstadt, Germany	SR-46349	4-((3Z)-3-(2-Dimethylaminoethyl)oxyimino-3-(2-fluorophenyl)propen-1-yl)phenol hemifumarate salt

## Data Availability

The original contributions presented in the study are included in the article/Appendix A; further inquiries can be directed to the corresponding author/s.

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
