# Peer review of "The Role of the Insular Cortex and Serotonergic System in the Modulation of Long-Lasting Nociception"

_cells, 2024, doi:10.3390/cells13201718_

Round 1

Reviewer 1 Report

Comments and Suggestions for Authors

The article is well written and highlights the the role of the serotoninergic system in the IC in long-lasting inflammatory pain.

There are a few minor comments pertaining to the figures and result section:

1. The result section can be sub-divided but there should be heading summarizing each findings.

2. Figure legends are not clear or intuitive in this article. Figure legends must define the content of the figures precisely. Please also do check the alignment of the graphs and blots with the legend (text paragraph)

3.The axis of the graphs in Fig. 2 are not legible.

Author Response

We would like to thank you for your comments. We address the points below.

Q1R1. The result section can be sub-divided but there should be heading summarizing each finding.

A1R1. We added subheadings summarizing the findings of each section of the results based on reviewer comments.

Q2R1. Figure legends are not clear or intuitive in this article. Figure legends must define the content of the figures precisely. Please also do check the alignment of the graphs and blots with the legend (text paragraph).

A2R1. According to his suggestion, the figure legends were rewritten to define the content of the figures more precisely.

Q3R1.The axis of the graphs in Fig. 2 are not legible.

A3R1. We apologize for this. In the new version, axis was corrected to be legible.

Reviewer 2 Report

Comments and Suggestions for Authors

The paper by Coffeen et al. describes extracellular 5HT and its metabolite 5HIAA, and the expression of 5HT receptors in the insula after  inflammation. The role of insular 5HT receptors in pain after inflammation is also investigated.  The topic is potentially very interesting, however the work presented is very rudimentary with some inaccuracies.

The authors only compare 3 groups for the microdialysis study: Control, Inflammation and Mechanociception, with the inflammation group also receiving mechanociception. Thus a pure inflammation, without mechanociception, control group should also be added.  

The term 'mechanociception' group, here, is misleading. It implies that the animals receive adequate mechanociceptive stimulation to influence the outcome of the microdialysis study. However, the animals are only tested for mechanical thresholds in two time points. And there seems not to be an obvious effect on 5HT/5HIAA as shown in the data. The authors should instead apply a superthreshold nociceptive stimulus repeatedly (every 30s for example) for a sufficient period of time (10-20 min) and analyze the data time dependently (baseline, before, during and after mechanociception) with  the appropriate controls.

In the western blot experiments the authors  find significance between one group and the others only when they removed the outliers. Although it is acceptable to remove outlines the issue here is that the n's, which were small to start with, is also reduced to unacceptable values (4). The authors should add more n's to the baseline group.

Pharmacological experiments were done only with carrageenan injected animals. Controls with vehicle instead of carrageenan should also be included

Are the data in figure 3 presented as SD or SEM? They dont look very significant. What are the n's for each group?

What are the total n's. Are the animals used in the different experiments different sets or are animals reused between experiments?

 Some of the statistics should be done with repeated measures ANOVA    

Comments on the Quality of English Language

average

Author Response

Thanks to the reviewer for his extensive and appropriate comments on our manuscript. Below, we respond point by point to all of his suggestions.

Q1R2. The authors only compare 3 groups for the microdialysis study: Control, Inflammation and Mechanociception, with the inflammation group also receiving mechanociception. Thus a pure inflammation, without mechanociception, control group should also be added.

A1R2. According to our results, acute nociceptive stimulation does not generate any change in the extracellular release of serotonin or its metabolite over a period of 180 minutes. We applied the first test to the 3 groups prior to the carrageenan injection, that is, prior to the start of the inflammatory process and we did not see any change with respect to baseline. Furthermore, repeated acute stimulation during the development of an inflammatory process does not induce any change in the tendency produced by the inflammation itself. This is like what we found in a previous study with another neurotransmission system (dopaminergic), in which acute stimulation (thermociception) does not produce any change in extracellular dopamine release (Coffeen et al., 2010, https:/ /doi.org/10.1186/1744-8069-6-75). We could interpret this as the inability of the acute stimulus to trigger a change in the release of these neurotransmission systems, possibly due to the nature of the response to acute stimulation, which occurs at the spinal level and not in the higher nuclei. And it neither enhances nor interferes with the development of the inflammatory process. Therefore, we do not consider it necessary to do another group, which probably will not provide us with different information in this regard.

Q2R2. The term 'mechanociception' group, here, is misleading. It implies that the animals receive adequate mechanociceptive stimulation to influence the outcome of the microdialysis study. However, the animals are only tested for mechanical thresholds in two time points. And there seems not to be an obvious effect on 5HT/5HIAA as shown in the data. The authors should instead apply a superthreshold nociceptive stimulus repeatedly (every 30s for example) for a sufficient period of time (10-20 min) and analyze the data time dependently (baseline, before, during and after mechanociception) with  the appropriate controls.

A2R2. We consider that our experimental design is adequate. Excessive repeated stimulation (every 30 seconds) could produce, on the one hand, habituation by the animal to the acute stimulus, and, on the other hand, it could cause tissue damage, so we would not be able to discern if the effect is present by repeated acute stimulation or by tissue damage itself. The tests we chose give us a good idea about acute stimulation. We applied the first test to the 3 groups prior to the carrageenan injection, that is, prior to the start of the inflammatory process; we applied the second during the first hour after the onset of inflammation; and we applied the third test 3 hours after the carrageenan injection, just when the nociceptive response given by the inflammation reaches its highest point.

Q3R2. In the western blot experiments the authors find significance between one group and the others only when they removed the outliers. Although it is acceptable to remove outlines the issue here is that the n's, which were small to start with, is also reduced to unacceptable values (4). The authors should add more n's to the baseline group.

A3R2. According to the reviewer's suggestions, in the new version of our manuscript, we incorporated five mice per group to have 11 subjects per group. The statistics were performed with all the subjects (N=33). Figure 2 shows the results of the 11 subjects per group. Original blots for the first 6 subjects remain in Figure 2. Complementary blots (experiments 1 and 2) are shown in supplementary figure 3. The results continue in the same direction, with a statistically significant difference for 5HT1A receptors. So, we believe the results are more plausible.

Q4R2. Pharmacological experiments were done only with carrageenan injected animals. Controls with vehicle instead of carrageenan should also be included.

A4R2. Our group has previously shown that intraplantar administration of vehicle (saline solution) does not produce any increase in the perimeter of the injected paw compared to the inflammation produced by carrageenan (López-Avila A. et al., 2004, doi: 10.1016/j.brainresprot.2004.01.001). Additionally, we have demonstrated that neither neuropathic nor inflammatory nociception is affected by this treatment. (Coffeen et al., 2009, Vol. 32 No. 2, Mental Health, https://revistasaludmental.ddns.net/index.php/salud_mental/article/view/1277). Therefore, we do not believe that conducting an additional group with the administration of the vehicle is necessary.

Q5R2. Are the data in figure 3 presented as SD or SEM? They dont look very significant. What are the n's for each group?

A5R2. Thank you for this observation. The data in figure 3 is presented as SEM. We updated this point to make it clear in the manuscript.

Although the effect is not as noticeable as for the thermonociception tests, there are significant differences in the group microinjected with the 5HT3 antagonist when compared with the control, 5HT, and 5HIAA groups at one hour, and it is only different with the 5HT group at 3 hours (*, p < 0.005).

The total n per group for this experimental series is stated in the material and methods section (lines 197 and 198). In any case, we added this information to the results section, and the figure legends were rewritten to make them much more descriptive.

Q6R2. What are the total n's. Are the animals used in the different experiments different sets or are animals reused between experiments?

A6R2 The total n differs for each experimental series. For example, for microdialysis experiments, it is n = 8 (this is indicated in lines 135 to 136), and for pharmacological experiments, it is n = 10 per group (lines 174 and 175). Regarding the western blot experiments, new samples were added with a number of 11 subjects per group. The animals used in each experimental series were different; this point was clarified in the manuscript.

Q7R2. Some of the statistics should be done with repeated measures ANOVA.

Thank you for your pertinent observation. Actually, a repeated measures ANOVA was used to assess the microdialysis experiment outcomes. We made the relevant corrections in the material and methods section. Apologies for the mistake.

Reviewer 3 Report

Comments and Suggestions for Authors

Comments and Suggestions for Authors

The study is interesting, the quality of English language is good, and the aim of the study is clear: investigate the role of the serotoninergic system in the insular cortex in long-lasting inflammatory pain in rats. Nonetheless, throughout the manuscript abbreviations “5HT, 5HIAA, 5HT1A, 5HT2A, 5HT3, etc.” should be changed to “5-HT, 5-HIAA, 5-HT1A, 5-HT2A, 5-HT3, etc.”, in accordance with the majority of bibliographic references in online databases.

Minor comments

INTRODUCTION

Line 37, there are too many references [4-12], it would be better to reduce them.

MATERIALS AND METHODS

Line 107, why did the Authors only use male rats and not female ones? Is there a particular reason?

Lines 139-140, what kind of HPLC-ECD system did the Authors use to quantify 5-HT and 5-HIAA concentrations?

Line 165, change “anti-5HT3a” with “anti-5-HT3A”.

Line 182, change “alloed to” with “allowed to”.

Table 1., the font table is not the same.

Line 245, change “Tukey post hoc test” with “Tukey’s post hoc test”.

RESULTS

Line 261, be careful to specify what PWT means the first time it is introduced in the main text: please, insert “paw withdrawal threshold” before PWT.

Lines 261-262, the r and P values of Pearson’s correlation are different from those reported in figure 1 legend. What are the correct values?

DISCUSSION

Lines 524, change “can exerts” with “can exert”.

Lines 554, modify “in duction” to “induction”.

REFERENCES

Reference #51, it is incomplete: please, add year (1998), Journal name (Kopf Carrier), vol. number (50), and page numbers (1-6).

FIGURES AND CAPTIONS

Figure 1, I would insert asterisks to highlight the statistical significance in panels A and B.

Figure 2, the letters A, B, C, and D from the panels are missing.

Figure 4 caption (line 396), change “5HT2a” with “5-HT2A”.

Comments on the Quality of English Language

The quality of English language is good.

Author Response

Thank you for your appropriate revision. Below, we respond to all of your comments.

Minor comments

INTRODUCTION

Q1R3. Line 37, there are too many references [4-12], it would be better to reduce them.

A1R3. We reduced the references and left only the most relevant ones.

MATERIALS AND METHODS

Q2R3. Line 107, why did the Authors only use male rats and not female ones? Is there a particular reason?

A2R3. Pain processing certainly differs between male and female animals (Mogil et al., 2024; doi: 10.1016/j.neubiorev.2024.105749). Although it may be interesting to know if serotonin within the IC plays the same role in females during inflammatory pain or if there is differential pain processing due to sex, it is not the objective of the present study. The answer to that question will require a full investigation. Furthermore, the present study aims to follow the same line of research that we have addressed in previous works, in which we have always used male rats (Coffeen et al., 2008, DOI: 10.1186/1744-8069-6-75; Coffeen et al., 2011, DOI:10.1016/j.ejpain.2010.06.007; Coffeen et al., 2018, DOI: 10.1002/ejp.1120).

Q3R3. Lines 139-140, what kind of HPLC-ECD system did the Authors use to quantify 5-HT and 5-HIAA concentrations?

A3R3. We used a Waters 2695 separation module and a Waters 2465 electrochemical detector. We added this information to the materials and methods section.

Q4R3. Line 165, change “anti-5HT3a” with “anti-5-HT3A”.

A4R3. Change was done.

Q5R3. Line 182, change “alloed to” with “allowed to”.

A5R3. We corrected it.

Q6R3. Table 1., the font table is not the same.

A5R3. We corrected it.

Q7R3. Line 245, change “Tukey post hoc test” with “Tukey’s post hoc test”.

A7R3. We corrected it.

RESULTS

Q8R3. Line 261, be careful to specify what PWT means the first time it is introduced in the main text: please, insert “paw withdrawal threshold” before PWT.

A8R3. Thank you. We inserted it.

Q9R3. Lines 261-262, the r and P values of Pearson’s correlation are different from those reported in figure 1 legend. What are the correct values?

A9R3. We apologize for this error. We have corrected the r and p in the figure legend.

DISCUSSION

Q10R3. Lines 524, change “can exerts” with “can exert”.

A10R3. Change was done.

Q11R3. Lines 554, modify “in duction” to “induction”.

A11R3. Change was done.

REFERENCES

Q12R3. Reference #51, it is incomplete: please, add year (1998), Journal name (Kopf Carrier), vol. number (50), and page numbers (1-6).

A12R3. Thank you. We added the information.

FIGURES AND CAPTIONS

Q13R3. Figure 1, I would insert asterisks to highlight the statistical significance in panels A and B.

A13R3. We inserted asteriks in panels A and B.

Q14R3. Figure 2, the letters A, B, C, and D from the panels are missing.

AR2. In the new version of our manuscript, we incorporated five mice per group to have 11 subjects per group. The statistics were performed with all the subjects (N=33). Figure 2 shows the results of the 11 subjects per group. Original blots for the first 6 subjects remain in Figure 2. Complementary blots (experiments 1 and 2) are shown in supplementary figure 3.

Q15R3. Figure 4 caption (line 396), change “5HT2a” with “5-HT2A”.

A15R3. Change was done.